# Serum brain-derived neurotrophic factor levels in type 2 diabetes mellitus patients and its association with cognitive impairment: A meta-analysis

**Wan-li He[1]⊙, Fei-xia Chang[1]⊙, Tao Wang[1], Bi-xia Sun[1], Rui-rong Chen[2], Lian-ping Zhao[3]\***

1 Department of Medical Imaging Center, Gansu Provincial Maternal and Child Care Hospital (Gansu Provincial Central Hospital), Lanzhou, Gansu, China, 2 Department of Medical Imaging Center, Gansu Provincial Maternal and Child Care Hospital, Lanzhou, Gansu, China, 3 Department of Radiology, Gansu Provincial Hospital, Lanzhou, Gansu, China

⊙ These authors contributed equally to this work.
\* zlpzxzz@163.com

**Data Availability Statement:** All relevant data are within the manuscript and its Supporting Information files.

## Abstract

### Objective

To compare the serum levels of brain-derived neurotrophic factor (BDNF) in type 2 diabetes mellitus (T2DM) patients with healthy controls (HC) and evaluate the BDNF levels in T2DM patients with/without cognitive impairment.

### Methods

PubMed, EMBASE, and the Cochrane Library databases were searched for the published English literature on BDNF in T2DM patients from inception to December 2022. The BDNF data in the T2DM and HC groups were extracted, and the study quality was evaluated using the Agency for Healthcare Research and Quality. A meta-analysis of the pooled data was conducted using Review Manager 5.3 and Stata 12.0 software.

### Results

A total of 18 English articles fulfilled with inclusion criteria. The standard mean difference of the serum BDNF level was significantly lower in T2DM than that in the HC group (SMD: -2.04, z = 11.19, $P$ <0.001). Besides, T2DM cognitive impairment group had a slightly lower serum BDNF level compared to the non-cognitive impairment group (SMD: -2.59, z = 1.87, $P$ = 0.06).

### Conclusion

BDNF might be involved in the neuropathophysiology of cerebral damage in T2DM, especially cognitive impairment in T2DM.

**Funding:** The author(s) received no specific funding for this work.

**Competing interests:** The authors have declared that no competing interests exist.

## 1. Introduction

Type 2 diabetes mellitus (T2DM) is a chronic systemic metabolic disorder seriously affecting human health, which is triggered by genetic predisposition and environmental factors [1]. International Diabetes Federation estimates that T2DM occurs in over 400 million people and it is one of the largest epidemics worldwide [2]. T2DM manifesting through fasting and post-prandial hyperglycemia can induce various life-threatening co-morbidities and complications such as diabetic neuropathy and diabetic nephropathy [3,4]. Cognitive dysfunction is an important complication observed in type T2DM patients [5]. In addition, T2DM is an important risk factor implicated in cognitive deficits except aging and neurodegenerative disorder [6]. T2DM patients have a greater decline in cognitive function than those without T2DM [7]. Besides, it is reported that T2DM accelerates brain aging and cognitive decline [8]. T2DM is significantly associated with an increased risk of dementia and a large portion of T2DM patients with cognitive impairment eventually progress to dementia [9,10], which may represent a consequence of brain-specific insulin resistance and impaired glucose regulation [11]. However, the pathophysiological mechanisms of cerebral impairment in T2DM remain elucidated.

Brain-derived neurotrophic factor (BDNF), a member of the neurotrophic family of proteins, is most widely distributed in the central nervous system (CNS) [12]. It plays an important role in protecting neurons and synaptic activity [13]. BDNF was released from the brain to peripheral circulation [14], and there is a correlation between BDNF in serum and CNS, providing an alternative measure of BDNF changes [15]. Alternation of BDNF is observed in the pathophysiological basis of many neurodegenerative and psychiatric disorders [16], including Alzheimer's disease and depression [17,18]. Furthermore, the serum BDNF is a useful biomarker for executive cognitive impairment in schizophrenia patients [19,20]. In addition, the BDNF Val66Met polymorphism may be a major factor in the susceptibility to cognitive impairment which affects the secretion of mature BDNF [21]. A meta-analysis suggests that BDNF Val66Met is associated with cognitive impairment in Parkinson's disease [22], confirming that BDNF is a risk factor for this disorder [23]. Furthermore, BDNF is related to the regulation of glucose levels [24]. Exogenous BDNF reduces blood glucose concentrations and glycated hemoglobin in obese diabetic mice [25], which is consistent with the finding that there was a positive correlation between BDNF and insulin sensitivity [26]. Previous studies have revealed the relationship between serum BDNF and diabetic conditions in T2DM patients with controversial results [27–34]. However, the precise role of BDNF in the development of T2DM patients as well as in cognitive function remains unclear.

Therefore, our study aims to explore the alteration tendency of the serum BDNF levels in T2DM patients with or without cognitive impairment using meta-analysis with a comprehensive evaluation of relevant literature. The current study will provide a basic foundation for further investigating the neuropathophysiological mechanisms of cerebral damage in T2DM.

## 2. Methods

### 2.1. Literature search and selection

A systemic search strategy was used to identify the relevant studies published in PubMed, EMBASE, and the Cochrane Library from inception to December 2022. We applied a search strategy based on the combination of relevant terms. Two independent investigators acquired articles and sequentially screened their titles and abstracts for eligibility. Then, full texts of articles deemed potentially eligible were acquired. Any disagreement would be solved via discussion with the help of a third senior investigator. A screening guide was used to ensure that the selection criteria were constantly applied.

Inclusion criteria: (1) clinical cross-sectional studies concerning the quantitative values of serum BDNF level in T2DM patients; (2) sufficient data were available for mean and standard deviation analysis of BDNF level; (3) original research. Exclusion criteria: (1) review, abstracts only, letters, comments, guidelines, and case reports; (2) studies *in vitro* or in animal models; (3) duplicate publications; (4) incomplete data.

## 2.2. Quality evaluation and data extraction

Agency for Healthcare Research and Quality (AHRQ) was used to evaluate the quality of the included cross-sectional studies. The AHRQ included 8 items with a total score of 8 points. Two independent researchers assessed the quality of the literature and reached a consensus after consultation when necessary.

The serum BDNF values (ng/ml) were extracted, and the indicators were represented by $\bar{x} \pm s$. Basic information includes the author of the document, year of publication, sample size, population distribution, country, diagnostic criteria, and measurement methods of BDNF. When standard deviations were unreported, they were calculated according to confidence interval (95% or 99%), or interquartile range or $P$-value. The calculation process was as follows:

Calculate standard deviation from confidence interval:

$SD = \sqrt{N}(Upper\ confidence\ interval - Lower\ confidence\ interval)/3.92(95\%)$

$SD = \sqrt{N}(Upper\ confidence\ interval - Lower\ confidence\ interval)/5.15(99\%)$

$SD = \sqrt{N}(Upper\ confidence\ interval - Lower\ confidence\ interval)/2t$ [t = tinv $(p,$ Degrees of freedom)];

(2) Calculate the standard deviation from an interquartile range:

$SD = (Quartile\ ceiling - Lower\ quartile)/1.35)$;

(3) Calculate standard deviation by $p$-value:

$SD = SE\sqrt{1/NE + 1/NC}$, SE = MD/t (MD : mean difference, NE : numbers of experimental, NC : numbers of control)

## 2.3. Statistical analysis

All the meta-analyses were performed on Review Manager 5.3 and STATA12.0 with a significance level of $P < 0.05$. To calculate the effect size for each study, the summary standard mean difference (SMD) and 95% confidence interval were applied to evaluate the serum BDNF values between T2DM and healthy control (HC), T2DM with or without cognitive impairment. Pooled SMD and corresponding 95% confidence interval were calculated using the inverse variances method. Heterogeneity was estimated using the Cochran Q ($P$) and the inconsistency index where a $P$ value less than 0.05 and $I^2$ value greater than 50% indicated the presence of significant heterogeneity across the enrolled studies. If notable heterogeneity was observed, a random-effect model was applied and subgroup analyses were used to determine factors that contributed to the heterogeneity and to explore how those factors influenced the results. Subgroup analysis was stratified by the BDNF measuring instruments brand (China or USA; same brand in China or USA), ethnicity (Asian or European), and population [adults or the aged (years≥60)]. In addition, sensitivity analysis was performed to evaluate the reliability of included studies using STATA 12.0. The Egger's test and the Begg's test were applied to evaluate potential publication bias using STATA 12.0.

## 3. Results

### 3.1. Search and selection results

The main search strategy is illustrated in **Table 1.** Studies selection was managed using End-Note X7. A total of 678 records were initially identified, but only 501 records remained after

**Table 1. Search strategy for identifying studies.**

| | Search strategy |
|---|---|
| #1 | ("Diabetes Mellitus, Type 2"[Mesh]) OR ("Diabetes Mellitus, Noninsulin-Dependent"[Title/Abstract]) OR ("Diabetes Mellitus, Ketosis-Resistant"[Title/Abstract]) OR ("Diabetes Mellitus, Ketosis Resistant"[Title/Abstract]) OR (Ketosis-Resistant Diabetes Mellitus[Title/Abstract]) OR ("Diabetes Mellitus, Non-Insulin Dependent"[Title/Abstract]) OR ("Diabetes Mellitus, Non-Insulin-Dependent"[Title/Abstract]) OR (Non-Insulin-Dependent Diabetes Mellitus[Title/Abstract]) OR ("Diabetes Mellitus, Stable"[Title/Abstract]) OR ("Stable Diabetes Mellitus"[Title/Abstract]) OR ("Diabetes Mellitus, Type II"[Title/Abstract]) OR (NIDDM [Title/Abstract]) OR ("Diabetes Mellitus, Noninsulin Dependent"[Title/Abstract]) OR ("Diabetes Mellitus, Maturity-Onset"[Title/Abstract]) OR ("Diabetes Mellitus, Maturity Onset"[Title/Abstract]) OR (Maturity-Onset Diabetes Mellitus[Title/Abstract]) OR (Maturity Onset Diabetes Mellitus[Title/Abstract]) OR (MODY [Title/Abstract]) OR ("Diabetes Mellitus, Slow-Onset"[Title/Abstract]) OR ("Diabetes Mellitus, Slow Onset"[Title/Abstract]) OR (Slow-Onset Diabetes Mellitus[Title/Abstract]) OR (Type 2 Diabetes Mellitus[Title/Abstract]) OR (Noninsulin-Dependent Diabetes Mellitus[Title/Abstract]) OR (Noninsulin Dependent Diabetes Mellitus[Title/Abstract]) OR (Maturity-Onset Diabetes[Title/Abstract]) OR ("Diabetes, Maturity-Onset"[Title/Abstract]) OR (Maturity Onset Diabetes[Title/Abstract]) OR (Type 2 Diabetes[Title/Abstract]) OR ("Diabetes, Type 2"[Title/Abstract]) OR ("Diabetes Mellitus, Adult-Onset"[Title/Abstract]) OR (Adult-Onset Diabetes Mellitus[Title/Abstract]) OR("Diabetes Mellitus, Adult Onset"[Title/Abstract]) |
| #2 | ("Brain-Derived Neurotrophic Factor"[Mesh]) OR (Brain-Derived Neurotrophic Factor [Title/Abstract]) OR ("Factor, Brain-Derived Neurotrophic"[Title/Abstract]) OR ("Neurotrophic Factor, Brain-Derived"[Title/Abstract]) OR (BDNF[Title/Abstract]) |
| #3 | #1 AND #2 |

the elimination of duplicates. Only 51 records were remaining after screening titles, and subsequently, 29 records remained after reading the abstract. After reading full texts, 11 articles with incomplete data were excluded and finally, 18 articles were enrolled. The flow diagram is shown in **Fig 1**.

## 3.2. Characteristics and quality evaluation

Eighteen articles were included in the meta-analysis. The basic characteristics and quality evaluation of the studies are shown in **Table 2**. Among them, 17 articles had T2DM and HC groups, and 3 articles divided the T2DM group into two subgroups according to the presence of cognitive impairment. Of the 18 articles included, 13 were done in China, 2 in Japan, and 1 in each of the following countries (USA, Italy, and Turkey). The sample's mean age was >18 years in 15 articles and >60 years in 3 articles. The diagnostic criteria of T2DM as recommended by the World Health Organization were adopted in 11 articles; whereas the American Diabetes Association was employed in 1 article, but the remaining articles were not mentioned. Measurement of BDNF using ELISA in 17 articles, but 1 article was not mentioned. All of the 18 included studies were cross-sectional studies. Based on the quality evaluation of AHRQ, 11 studies scored 8, 4 studies scored 7, 1 study scored 5, and 1 study scored 4.

## 3.3. Meta-analysis

We compared the BDNF level between T2DM and HC groups ($P < 0.001$, $I^2 = 99\%$), and between the T2DM with or without cognitive impairment groups ($P < 0.001$, $I^2 = 90\%$) using a random-effect model since the heterogeneity test showed the $I^2$ value >50%.

Seventeen articles contained 2966 T2DM cases and 3580 HCs. The serum BDNF level in the T2DM group was significantly lower than that in the HC group [SMD: -2.04, z = 11.19, $P < 0.001$] (**Fig 2A**). The number of T2DM patients with or without cognitive impairment was 672 and 1913, respectively. The serum BDNF levels in T2DM with cognitive impairment group had a marginal difference from those without cognitive impairment [SMD: -2.59, z = 1.87, $P = 0.06$] (**Fig 2B**).

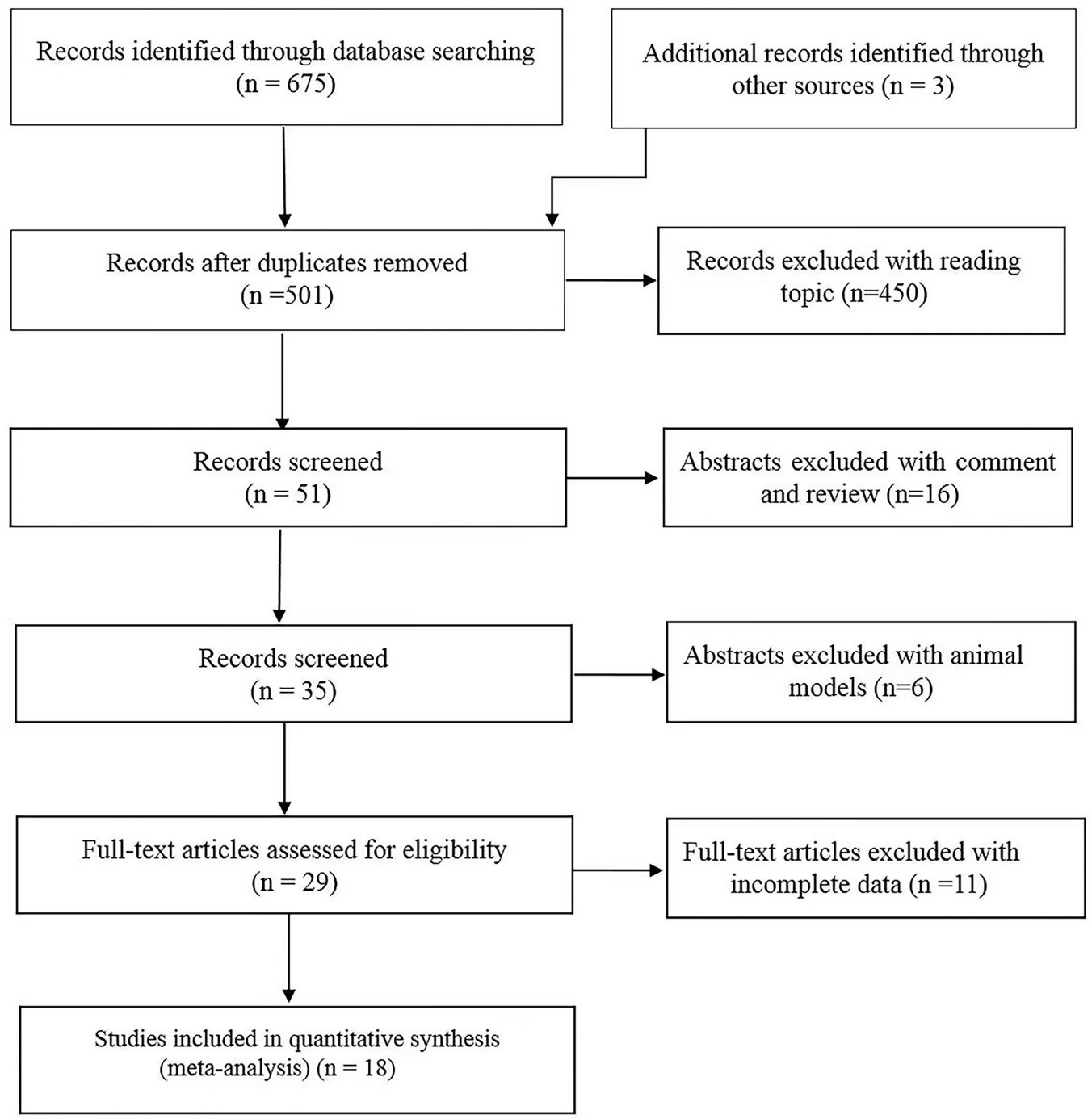

**Fig 1. Flow diagram of the study.**

### 3.4. Sensitivity analysis

Sensitivity analyses were conducted to evaluate the robustness of the findings by excluding 1 study at a time to assess if the results were driven by any one study. The significance of the meta-analysis outcome for T2DM and HC group changed when ruling out any one of 6/17 studies and the results also changed in T2DM with or without cognitive impairment group after ruling out 1/3 study, suggesting the results were unstable.

**Table 2. Basic characteristics of literature and literature quality evaluation form.**

| Study and Year | Research Subject | | Population | Country | Diagnostic Criteria of T2DM | Measurement of BDNF | Type of Study | AHRQ (points) |
|---|---|---|---|---|---|---|---|---|
| | T2DM (CI/NCI) | HC | | | | | | |
| Yan-Feng Zhen 2019 [27] | 230 | 248 | Adult | China | WHO | ELISA | Cross-sectional | 8 |
| Zhi-Chun Sun 2018 [28] | 715(151/564) | 396 | Adult | China | WHO | ELISA | Cross-sectional | 8 |
| Yan-Feng Zhen 2019 [27] | 311 | 346 | Adult | China | WHO | ELISA | Cross-sectional | 8 |
| Nicole.L.Spartano 2019 [35] | 179 | 1551 | Adult | USA | WHO | ELISA | Cross-sectional | 8 |
| Qin Sun 2018 [36] | 83 | 110 | Adult | China | WHO | ELISA | Cross-sectional | 8 |
| Tian Peng Zheng 2018 [37] | 1833(502/1331) | N | The aged | China | WHO | ELISA | Cross-sectional | 8 |
| Qing-Guo Ren 2017 [38] | 89 | 40 | The aged | China | Unclear | ELISA | Cross-sectional | 7 |
| Min-Guo 2017 [30] | 404 | 212 | Adult | China | WHO | ELISA | Cross-sectional | 7 |
| Wei Liu 2016 [39] | 28 | 85 | Adult | China | Unclear | ELISA | Cross-sectional | 7 |
| Blanca Murillo Ortiz 2016 [32] | 37(19 /18) | 40 | Adult | Mexico | Unclear | ELISA | Cross-sectional | 5 |
| Huli Wei 2017 [40] | 92 | 81 | Adult | China | WHO | ELISA | Cross-sectional | 8 |
| Bo Li 2016 [31] | 292 | 200 | Adult | China | WHO | ELISA | Cross-sectional | 8 |
| Ming-He 2014 [29] | 37 | 37 | Adult | China | Unclear | Unclear | Cross-sectional | 4 |
| Banu Boyuk 2014 [41] | 88 | 33 | Adult | Turkey | Unclear | ELISA | Cross-sectional | 7 |
| Angela Passaro 2014 [42] | 37 | 122 | The aged | Italy | ADA | ELISA | Cross-sectional | 8 |
| Yan-Feng Zhen 2019 [27] | 208 | 212 | Adult | China | WHO | ELISA | Cross-sectional | 8 |
| Aya Fujinami 2008 [43] | 122 | 80 | Adult | Japan | WHO | ELISA | Cross-sectional | 8 |
| Masataka Suwa 2006 [34] | 24 | 7 | Adult | Japan | Unclear | ELISA | Cross-sectional | 6 |

**Abbreviations:** ADA, American Diabetes Association; WHO, World Health Organization; ELISA, Enzyme-Linked Immunosorbent Assay; T2DM, Type 2 Diabetes Mellitus; HC, Healthy Control; CI, Cognitive Impairment; NCI, Non-Cognitive Impairment.

## 3.5. Subgroup analysis

Subgroup analysis based on the BDNF measuring instruments (either China or USA) exhibited that there were significant differences in BDNF values between T2DM and HC (China: $P = 0.05$; USA: $P < 0.001$; Total: $P < 0.001$), with large heterogeneity (China: $P < 0.001$ and $I^2 = 100\%$; USA: $P < 0.001$ and $I^2 = 99\%$; Total: $P < 0.001$ and $I^2 = 99\%$) (**Fig 3**). Then, subgroup analysis was performed on the same instrument brand in China or the USA, respectively and similar results were observed (China: $P < 0.001$; USA: $P = 0.002$; Total: $P < 0.001$). The heterogeneity was only observed in the same brand from the USA, but not in the same brand from China [China: $(P = 0.84$ and $I^2 = 0\%$; USA: $P < 0.001$ and $I^2 = 98\%$; Total: $P < 0.001$ and $I^2 = 96\%$)] (**Fig 4**). Subgroup analysis based on ethnicity and population distribution presented

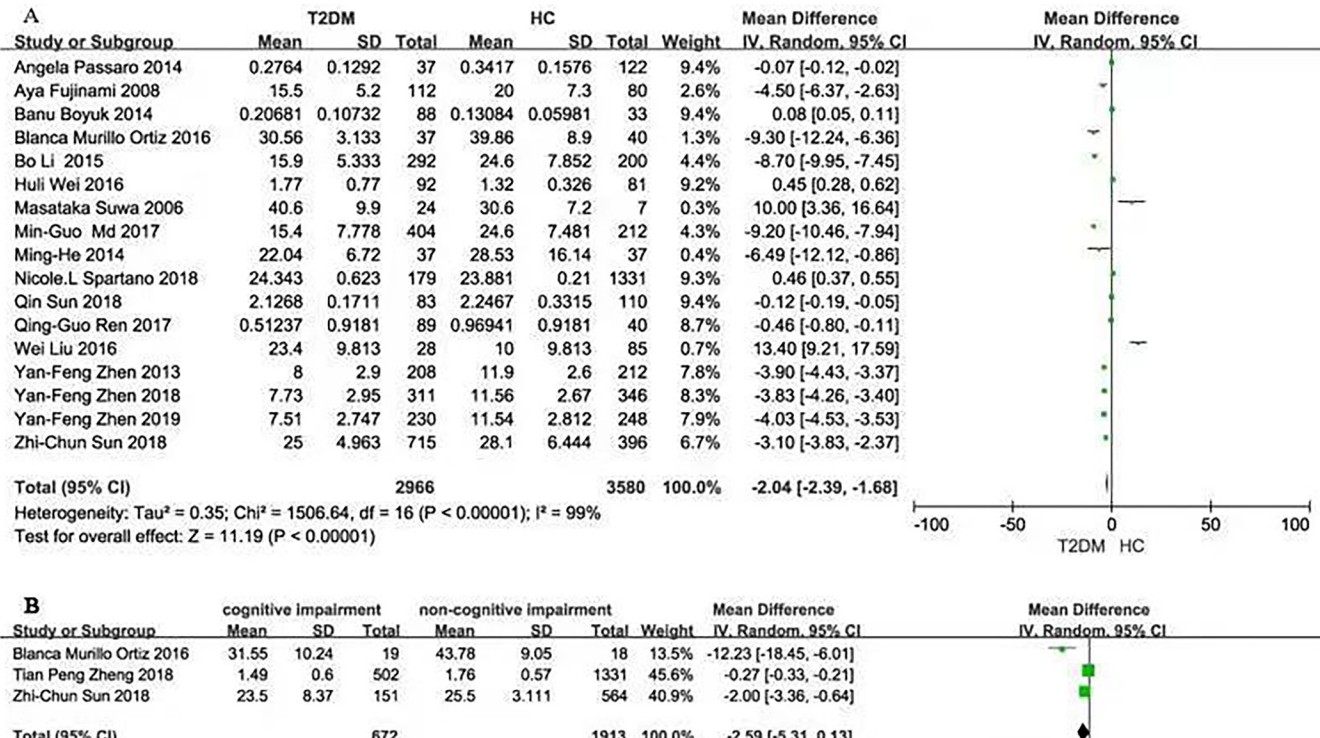

**Fig 2. Forest plot showing a standard mean difference for the BDNF levels. (A)** The different BDNF levels between T2DM and HC. **(B)** The different BDNF levels in T2DM patients with or without cognitive impairment. **Abbreviations:** BDNF, brain-derived neurotrophic factor; T2DM, type 2 diabetes mellitus; HC, healthy controls.

that the BDNF values were significantly different, except for the European (Asian: $P < 0.001$; European: $P = 0.59$; Total: $P < 0.001$). In addition, there was significant difference in the adults in T2DM and HC, except for the aged (adults: $P < 0.001$; the aged: $P = 0.25$; Total: $P < 0.001$), and there were no marked decrease in heterogeneity (Asian: $P < 0.001$ and $I^2 = 99\%$; European: $P < 0.001$ and $I^2 = 99\%$; Total: $P < 0.001$ and $I^2 = 99\%$) (Adults: $P < 0.001$ and $I^2 = 99\%$; the aged: $P = 0.03$ and $I^2 = 80\%$; Total: $P < 0.001$ and $I^2 = 99\%$) (**Figs 5 and 6**).

### 3.6. Publication bias analysis

After Egger's and Begg's test, the studies of T2DM and HC group, T2DM with or without cognitive impairment group showed no significant publication bias ($P = 0.606$, $P = 0.672$; $P = 0.202$, $P = 1.000$).

## 4. Discussion

Our study is the first meta-analysis to evaluate the levels of serum BDNF in T2DM patients and HCs and compare the levels between T2DM patients with or without cognitive impairment. We found that the serum BDNF levels were lower in T2DM compared with HC. Furthermore, the serum BDNF levels had a decreasing tendency in T2DM patients with cognitive impairment compared with those without cognitive impairment.

The BDNF plays a key role in the pathophysiology of T2DM due to improving glucose metabolism and insulin sensitivity [44–46]. Previous studies have reported that T2DM patients

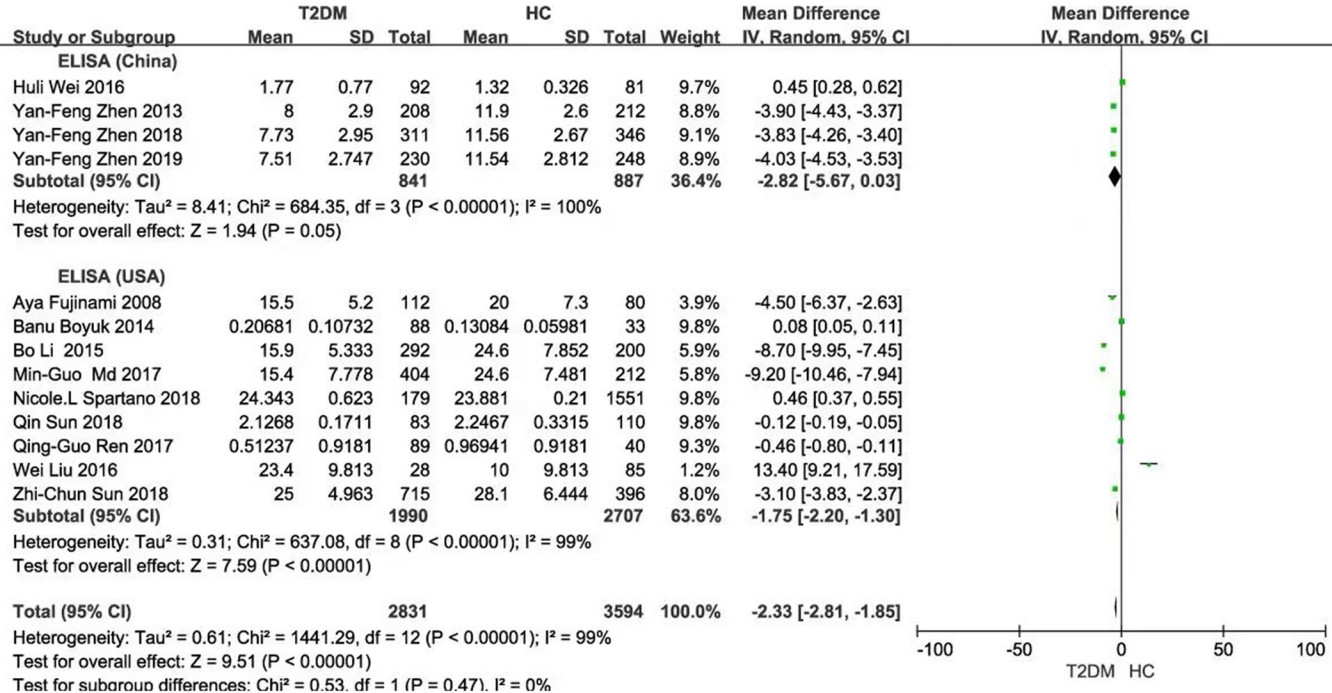

**Fig 3. Forest plot of subgroup analysis stratified by the BDNF measuring instruments brand China or USA.**

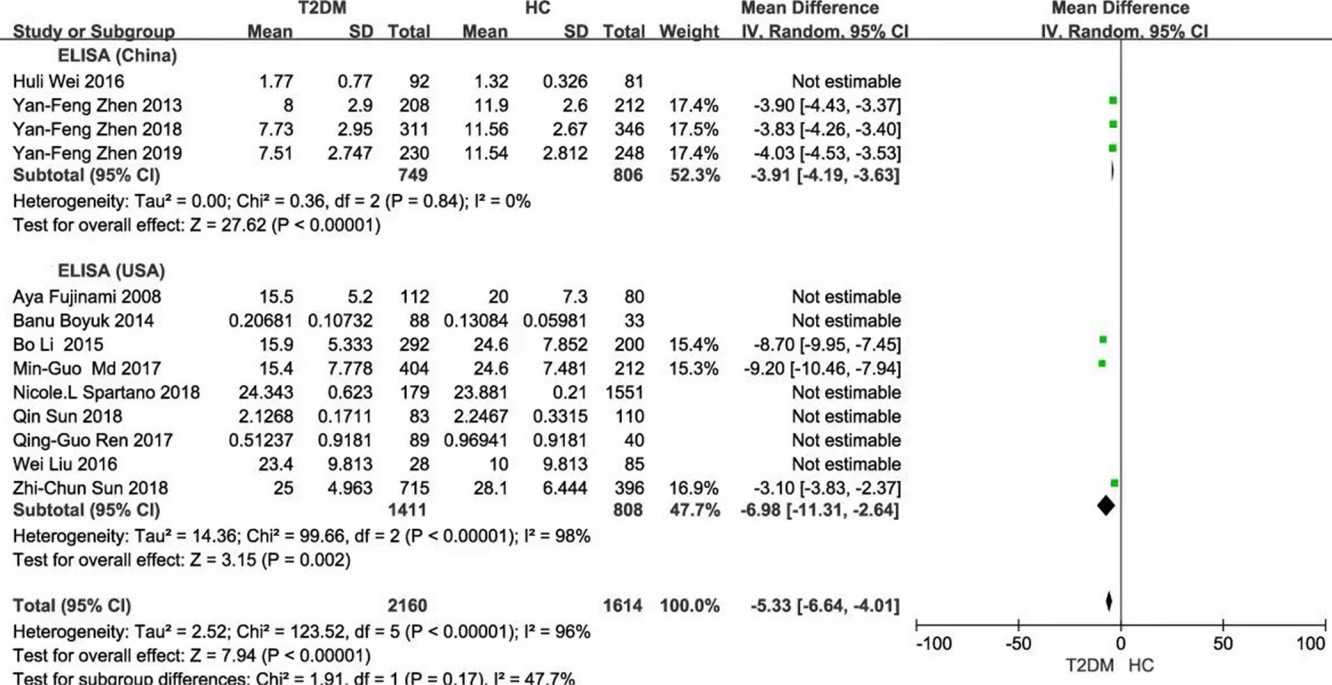

**Fig 4. Forest plot of subgroup analysis stratified by the same brand of instrument from China or the USA.**

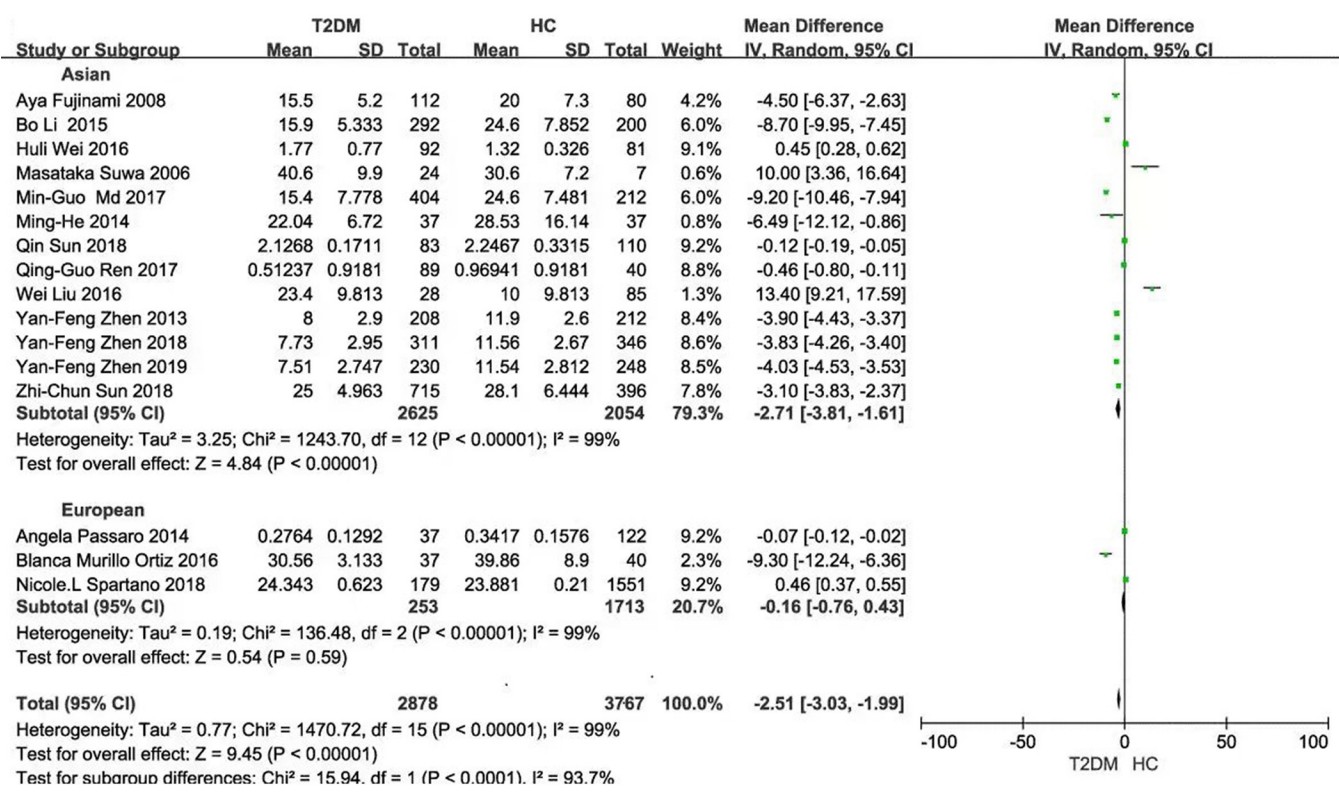

**Fig 5. Forest plot of subgroup analysis stratified by the ethnicity Asian or European.**

exhibited significantly lower levels of serum BDNF compared with normal controls [27–32,43], which is consistent with our research. Additionally, the cerebral output of BDNF is inhibited under hyperglycemia, logically decreased serum BDNF may be detected in the uncontrolled T2DM patients [14]. This is in line with the findings that there is an inverse correlation between serum BDNF levels and long-standing diabetes, in males and aged T2DM patients [43]. Interestingly, upregulated serum BDNF levels in T2DM patients were also reported [33,34]. Such discrepancy is possibly related to physical exercise, obesity, and a balanced diet in T2DM patients [47–50]. In addition, the serum BDNF levels are increased in T2DM patients who received metformin treatment [3]. This may also link to a compensatory mechanism of serum BDNF release in T2DM [33], which is supported by the findings that the upregulated serum BDNF levels control blood glucose in newly diagnosed T2DM patients, but this control ability might be lost in a long term T2DM patients [35]. Our explanation is further supported by a resting state fMRI report showing enhanced functional connectivity of the left hippocampus (a major source of BDNF) with the left inferior frontal gyrus in the early stage of T2DM, which might contribute to adaptive compensation of hippocampal function [51]. Taken together, the serum BDNF could be a useful biological marker to monitor the development of T2DM and the cerebral impairment in T2DM.

T2DM has reduced the number of new neurons in the hippocampus, and hippocampal neurogenesis plays an important role in learning and memory function throughout life [52]. The Hippocampal perhaps regulates BDNF to provide neuroprotection and control of synaptic interactions [53–56]. In the present meta-analysis, the serum BDNF levels presented a decreasing tendency in T2DM patients with cognitive impairment compared with those patients without cognitive impairment. Such downregulation was also observed in Alzheimer's disease, showing that the serum BDNF levels may be involved in the progression of cognitive

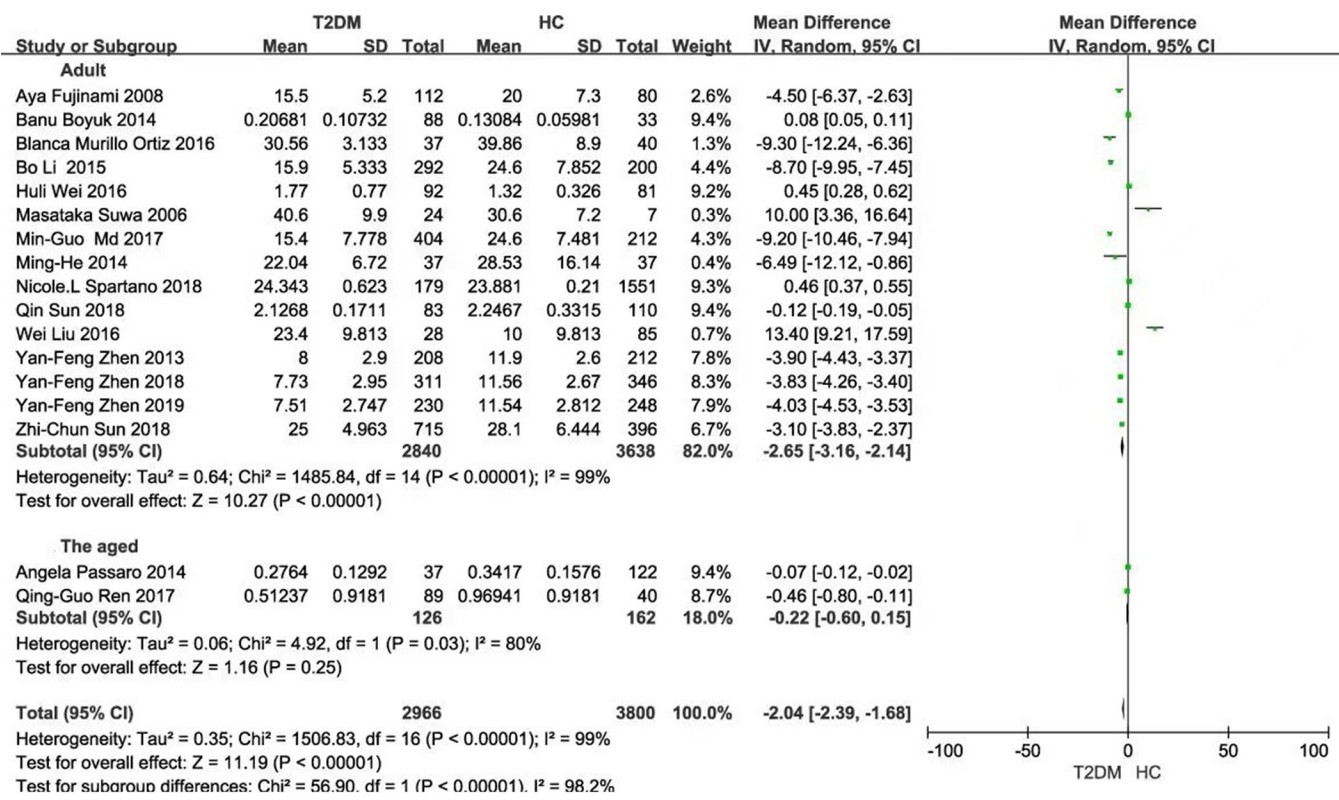

**Fig 6. Forest plot of subgroup analysis stratified by the population adults or the aged.**

impairment [57]. Such findings showed that serum BDNF levels may be involved in the progression of cognitive impairment in patients with T2DM. Thus, the reduction of BDNF might contribute to the neuropathophysiology of brain damage in T2DM, especially relating to cognitive impairment in T2DM.

However, substantial heterogeneity existed in the present meta-analysis. The heterogeneity could be generated from related factors, including the different brands of instruments for measuring BDNF, times and methods of blood collection, population distributions, and ethnicities. Such heterogeneity has been eliminated in the subgroup analysis by comparing the data from the same brands of instruments.

There are some limitations in the study. Firstly, the meta-analysis mostly included Chinese Han populations, which may not reflect the entire population/race. Secondly, different diagnostic criteria for diabetes were applied which might also compromise the data analysis. Although internationally recognized scales were utilized, the lack of a standard protocol for cognitive impairment could lead to inconsistent results.

In conclusion, the present meta-analysis suggests that the decrease in serum BDNF levels in T2DM patients has resolved the inconsistencies in previous studies. The serum BDNF levels in T2DM patients with cognitive impairment had a downward trend compared with those patients without cognitive impairment. Moreover, the reduction of serum BDNF may be a vital neuropathophysiological mechanism of cognitive impairment in T2DM patients.

## Supporting information

**S1 Raw data.**
(XLSX)

**S2 Raw data.**
(XLS)

## Author Contributions

**Conceptualization:** Wan-li He, Fei-xia Chang.

**Data curation:** Wan-li He, Fei-xia Chang, Tao Wang, Lian-ping Zhao.

**Formal analysis:** Wan-li He, Tao Wang.

**Investigation:** Fei-xia Chang, Tao Wang, Bi-xia Sun.

**Methodology:** Wan-li He, Fei-xia Chang, Bi-xia Sun, Rui-rong Chen.

**Project administration:** Lian-ping Zhao.

**Visualization:** Bi-xia Sun, Rui-rong Chen, Lian-ping Zhao.

**Writing – original draft:** Wan-li He, Fei-xia Chang, Tao Wang, Bi-xia Sun, Rui-rong Chen, Lian-ping Zhao.

**Writing – review & editing:** Tao Wang, Bi-xia Sun, Rui-rong Chen, Lian-ping Zhao.

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
