## [Decision Letter · Decision Letter 0]

5 Nov 2023

PONE-D-23-23408Serum brain-derived neurotrophic factor levels in type 2 diabetes mellitus patients and its association with cognitive impairment: a meta-analysisPLOS ONE

Dear Dr. Zhao,

Thank you for submitting your manuscript to PLOS ONE. After careful consideration, we feel that it has merit but does not fully meet PLOS ONE’s publication criteria as it currently stands. Therefore, we invite you to submit a revised version of the manuscript that addresses the points raised during the review process.

We look forward to receiving your revised manuscript.

Kind regards,

Prof Purvi Purohit

Academic Editor

PLOS ONE

Additional Editor Comments:

The authors have discussed "Serum brain-derived neurotrophic factor levels in type 2 diabetes mellitus patients and

its association with cognitive impairment". The methodology is appropriate, and the results have been described in line with the findings. The manuscript does have merit for publication but is not suitable for it in its current state. In general, numerous grammatical errors and incomplete or unclear sentence structure hinder a lucid reading of the manuscript. A thorough proofread is recommended. The other issues are as follows:

Introduction

"T2DM is the most important risk factor implicated in cognitive deficits, including learning, memory, and

and processing speed."

Is T2DM the most important risk factor for cognitive deficit? What about aging and neurodegenerative disorders?

The subsection under "Introduction" pointing towards the connection between T2DM and cognitive deficits needs reframing and updating with recent literature. I do not see any literature post-2020 either in the background forming the research question or in the discussion of the results, even though a quick search on Pubmed or Google turns up plenty of recent research.

Methods

I suggest an update of the literature search for the meta-analysis, which is currently up to December 2022. Was the study registered on a database (e.g., Prospero)? If yes, it would be better to provide the ID.

"We applied a search strategy based on the combination of relevant terms. Two independent investigators acquired articles and sequentially screened their titles and abstracts for eligibility. Then, full texts of articles deemed potentially eligible were acquired."

How were the articles acquired and screened? Manually or through some screening tool?

" = √( − )/2 [t=tinv (p, Degrees of freedom)]"

The formula is not clear. What is p and what does tinv stand for?

Discussion

It is unclear whether the authors want to discuss BDNF in relation to T2DM-related cognitive impairment or cognitive impairment in general. The methodology certainly seems to point towards the former, but the "discussion" also has reference to Alzheimer's and "T2DM-comorbid depression", which is confusing. More supporting literature in favor of BDNF and T2DM-related cognitive impairment can be added.

Reviewers' comments:

Reviewer's Responses to Questions

**Comments to the Author**

1. Is the manuscript technically sound, and do the data support the conclusions?

Reviewer #1: Partly

2. Has the statistical analysis been performed appropriately and rigorously? 

Reviewer #1: Yes

3. Have the authors made all data underlying the findings in their manuscript fully available?

Reviewer #1: No

4. Is the manuscript presented in an intelligible fashion and written in standard English?

Reviewer #1: No

5. Review Comments to the Author

Reviewer #1: The authors have discussed "Serum brain-derived neurotrophic factor levels in type 2 diabetes mellitus patients and

its association with cognitive impairment". The methodology is appropriate, and the results have been described in line with the findings. The manuscript does have merit for publication but is not suitable for it in its current state. In general, numerous grammatical errors and incomplete or unclear sentence structure hinder a lucid reading of the manuscript. A thorough proofread is recommended. The other issues are as follows:

Introduction

"T2DM is the most important risk factor implicated in cognitive deficits, including learning, memory, and

and processing speed."

Is T2DM the most important risk factor for cognitive deficit? What about aging and neurodegenerative disorders?

The subsection under "Introduction" pointing towards the connection between T2DM and cognitive deficits needs reframing and updating with recent literature. I do not see any literature post-2020 either in the background forming the research question or in the discussion of the results, even though a quick search on Pubmed or Google turns up plenty of recent research.

Methods

I suggest an update of the literature search for the meta-analysis, which is currently up to December 2022. Was the study registered on a database (e.g., Prospero)? If yes, it would be better to provide the ID.

"We applied a search strategy based on the combination of relevant terms. Two independent investigators acquired articles and sequentially screened their titles and abstracts for eligibility. Then, full texts of articles deemed potentially eligible were acquired."

How were the articles acquired and screened? Manually or through some screening tool?

" = √( − )/2 [t=tinv (p, Degrees of freedom)]"

The formula is not clear. What is p and what does tinv stand for?

Discussion

It is unclear whether the authors want to discuss BDNF in relation to T2DM-related cognitive impairment or cognitive impairment in general. The methodology certainly seems to point towards the former, but the "discussion" also has reference to Alzheimer's and "T2DM-comorbid depression", which is confusing. More supporting literature in favor of BDNF and T2DM-related cognitive impairment can be added.

6. PLOS authors have the option to publish the peer review history of their article (what does this mean?). If published, this will include your full peer review and any attached files.

Reviewer #1: **Yes: **Dipayan Roy

---

## [Author Response · Author response to Decision Letter 0]

10 Dec 2023

Response letter to reviewers

Dear Editors and Reviewers:

Thank you for your letter and for the reviewers’ comments concerning our manuscript entitled “Serum brain-derived neurotrophic factor levels in type 2 diabetes mellitus patients and its association with cognitive impairment: a meta-analysis” (Manuscript Number.: PONE-D-23-23408). Those comments are all valuable and very helpful for revising and improving our paper, as well as the important guiding significance to our research. We have studied the comments carefully and have made corrections which we hope meet with approval. Revised portions are marked in red on the paper. The main corrections in the paper and the responses to the reviewer’s comments are as follows:

Reviewer #1: The authors have discussed "Serum brain-derived neurotrophic factor levels in type 2 diabetes mellitus patients and

its association with cognitive impairment". The methodology is appropriate, and the results have been described in line with the findings. The manuscript does have merit for publication but is not suitable for it in its current state. A thorough proofreading is recommended. The other issues are as follows:

Response: Thanks for your time and effort spent reviewing our manuscript. We have checked our manuscript thoroughly and made relevant corrections to improve paper quality. 

Introduction

"T2DM is the most important risk factor implicated in cognitive deficits, including learning, memory, and processing speed. "Is T2DM the most important risk factor for the cognitive deficit? What about aging and neurodegenerative disorders?

Response: Thank you for your comments. “T2DM is the most important risk factor for cognitive deficit except aging and neurodegenerative disorders.” We have made relevant corrections in the Introduction section. (Introduction section, paragraph 1) 

The subsection under "Introduction" pointing towards the connection between T2DM and cognitive deficits needs reframing and updating with recent literature. I do not see any literature post-2020 either in the background forming the research question or in the discussion of the results, even though a quick search on Pubmed or Google turns up plenty of recent research.

Response: Thank you for your comments. We have added literature post-2020 in the Introduction and Discussion.

Methods

I suggest an update of the literature search for the meta-analysis, which is currently up to December 2022. 

Response: Thank you for your comments. The literature as of October 2022 has been retrieved but no other relevant literature has been published.

Was the study registered on a database (e.g., Prospero)? If yes, it would be better to provide the ID.

Response: Thank you for your comments. The study was registered on a database and

the ID was CRD42020211608.

"We applied a search strategy based on the combination of relevant terms. Two independent investigators acquired articles and sequentially screened their titles and abstracts for eligibility. Then, full texts of articles deemed potentially eligible were acquired."

How were the articles acquired and screened? Manually or through some screening tool?

Response: Thank you for your comments. A systemic search strategy was used to identify the relevant studies published in PubMed, EMBASE, and the Cochrane Library from inception to December 2022. We applied a search strategy based on the combination of relevant terms. Two independent investigators acquired articles and sequentially screened their titles and abstracts for eligibility. Then, full texts of articles deemed potentially eligible were acquired. Any disagreement would be solved via discussion with the help of a third senior investigator. A screening guide was used to ensure that the selection criteria were constantly applied. The articles were acquired and screened manually. 

Inclusion criteria: (1) clinical cross-sectional studies concerning the quantitative values of serum BDNF level in T2DM patients; (2) sufficient data were available for mean and standard deviation analysis of BDNF level; (3) original research. Exclusion criteria: (1) review, abstracts only, letters, comments, guidelines, and case reports; (2) studies in vitro or in animal models; (3) duplicate publications; (4) incomplete data.

" = √( − )/2 [t=tinv (p, Degrees of freedom)]"

The formula is not clear. What is p and what does tinv stand for?

Response: Thank you for your comments. p stands for probability and tinv is a function symbol.

Discussion

It is unclear whether the authors want to discuss BDNF in relation to T2DM-related cognitive impairment or cognitive impairment in general. The methodology certainly seems to point towards the former, but the "discussion" also has reference to Alzheimer's and "T2DM-comorbid depression", which is confusing. More supporting literature in favor of BDNF and T2DM-related cognitive impairment can be added.

Response: Thank you for your comments. We made some changes to the article.

---

## [Editor Report · Decision Letter 1]

12 Jan 2024

Serum brain-derived neurotrophic factor levels in type 2 diabetes mellitus patients and its association with cognitive impairment: a meta-analysis

PONE-D-23-23408R1

Dear Dr. Lian-ping Zhao,

We’re pleased to inform you that your manuscript has been judged scientifically suitable for publication and will be formally accepted for publication once it meets all outstanding technical requirements.

Kind regards,

Purvi Purohit

Academic Editor

PLOS ONE

Additional Editor Comments (optional):

Thank you for the revision. The manuscript is suitable for publication.

---

## [Editor Report · Acceptance letter]

25 Jan 2024

PONE-D-23-23408R1 

PLOS ONE

Dear Dr. Zhao, 

I'm pleased to inform you that your manuscript has been deemed suitable for publication in PLOS ONE. Congratulations! Your manuscript is now being handed over to our production team.

Kind regards, 

on behalf of

Dr. Purvi Purohit 

Academic Editor

PLOS ONE